# Highland Barley Starch: Structures, Properties, and Applications

**DOI:** 10.3390/foods12020387

**Published:** 2023-01-13

**Authors:** Jingjing Xie, Yan Hong, Zhengbiao Gu, Li Cheng, Zhaofeng Li, Caiming Li, Xiaofeng Ban

**Affiliations:** 1School of Food Science and Technology, Jiangnan University, Wuxi 214122, China; 2Key Laboratory of Synthetic and Biological Colloids, Ministry of Education, Wuxi 214122, China; 3Collaborative Innovation Center for Food Safety and Quality Control, Jiangnan University, Wuxi 214122, China; 4Qingdao Special Food Research Institute, Qingdao 266109, China

**Keywords:** highland barley starch, extraction, structure, properties, application

## Abstract

Highland barley (HB) is a nutritious crop with excellent health benefits, and shows promise as an economically important crop with diverse applications. Starch is the main component of HB and has great application potential owing to its unique structural and functional properties. This review details the latest status of research on the isolation, chemical composition, structure, properties, and applications of highland barley starch (HBS). Suggestions regarding how to better comprehend and utilize starches are proposed. The amylopectin content of HBS ranged from 74% to 78%, and can reach 100% in some varieties. Milling and air classification of barley, followed by wet extraction, can yield high-purity HBS. The surface of HBS granules is smooth, and most are oval and disc-shaped. Normal, waxy, and high-amylose HBS have an A-type crystalline. Due to its superb freeze-thaw stability, outstanding stability, and high solubility, HBS is widely used in the food and non-food industries. The digestibility of starch in different HB whole grain products varies widely. Therefore, the suitable HB variety can be selected to achieve the desired glycemic index. Further physicochemical modifications can be applied to expand the variability in starch structures and properties. The findings provide a thorough reference for future research on the utilization of HBS.

## 1. Introduction

Highland barley (HB; Hordeum vulgare L. var. nudum Hook. f.) is a variety of barley that belongs to the grass family and is an annual herb, which can be classified into two-, four-, and six-row HB based on the number of ridges, as well as white, black, and purple HB based on color. The characteristics of HB include a high degree of cold tolerance, a short growing period, wide adaptability, early maturity, and high yield, and it is suitable for cultivation in the cool climate of the plateau [1]. In 2019, Tibet and Qinghai, the major HB planting areas, produced 792,900 and 144,100 tons of HB, respectively, accounting for over 80% of total HB production.

In comparison with other cereal crops, HB had better nutritional value as it contains higher protein, vitamin, and fiber contents, especially β-glucan (Table 1). β-glucan serves a variety of physiological roles, such as reducing blood glucose and fat levels, lowering cholesterol, preventing colon cancer, and boosting immunity [2]. Owing to its rich nutritional value and unique taste and flavor, HB has been processed and formulated into many different types of food products, including noodles [3], bread [4], biscuits [5], cakes [6], vinegar [7], and wine [8]. 

The primary component of HB is starch, which accounts for 58.1–72.2% of the dry weight and generally comprises 74–78% amylopectin, and up to 100% in a few varieties [9]. The structures and properties of starch have a significant impact on the quality of HB products. For example, a higher content of amylopectin enables HB flour to have better freeze-thaw stability and can be added to other flours to enhance the quality of noodle products. These quality characteristics are key factors affecting the processing of HB noodle products. Therefore, a better understanding of the structure and functionality of HBS may help expand the application of this starch in food and other industries.

In comparison to other common starches, HBS has not been analyzed systematically, which hinders the in-depth research of HB and the application of HBS. In this review, we aim to summarize the isolation methods, chemical composition, structure, properties, and applications of HBS. Additionally, future research suggestions are recommended.

## 2. Isolation

Physical techniques, enzymatic hydrolysis, and other procedures can be used to decompose the cell wall of HBS, which consists of cellulose, hemicellulose, and pectin. Following the release of the cellular contents, proteins can be separated using appropriate procedures to yield pure HBS. The current methods for extracting HBS are the dry, wet, and wet-dry combination approaches.

### 2.1. Dry Extraction

Dry extraction is a useful process for enriching certain nutrients (starch, protein, lipids, and β-glucan) in barley. Pearling [23,24], roller milling [25,26], milling followed by air classification [27,28], and milling followed by sieving [29,30] are some of the dry fractionation procedures reported by researchers. Liu et al. [30] stated that pearling has a considerable impact on the efficiency of subsequent milling procedures. The milling process and the barley genotype had a substantial influence on the effectiveness of sieving for nutrient enrichment and recovery rates. Pearling alone was the optimum strategy for enriching protein, whereas, for β-glucan and starch enrichment, a combination of pearling and milling followed by sieving was the optimal choice. Compared to wet extraction, dry fractionation uses less energy and water and preserves the natural structure and function of the components. However, these methods are currently not capable of producing high-purity isolates (>90%) [31]. It is noteworthy that during the milling process, the starch granules are inflicted to various forces, which cause them to break into smaller particles, called milling damaged starch (MDS) [32]. Due to its unique structure, MDS has a considerable impact on the quality of the final starchy products. For example, MDS has higher water absorption capacity and enzymatic hydrolysis rate, and is easily fermented by yeasts. During bread making, the suitable amount of starch can improve the quality of the dough, while excessive starch can lead to sticky dough [33,34,35].

### 2.2. Wet Extraction

Wet extraction of HBS frequently requires the use of alkaline and enzymatic techniques. Soaking with lye can degrade or loosen the protein around the HBS, weaken the HBS-protein combination, and dissolve the protein to obtain high-purity starch. Yang et al. [36] extracted HBS by soaking the grains in NaOH solution using a 1:8 ratio at 30 °C for 8 h. Although the alkaline extraction is straightforward, multiple washes, centrifugation, acid neutralization following alkaline washing, wastewater treatment, and desalination are components of the alkaline extraction technique and are time-consuming. HBS extracted by the alkali method is characterized by its low gelatinization temperature, poor thermal stability, low degree of retrogradation, and a relatively smooth surface [37].

The enzyme cellulase is commonly used for starch extraction, and it acts on the cell wall of HB to rupture and disintegrate it, enabling the complete release of cellular contents, which is beneficial for the separation of starch and protein [38]. Extraction of highland barley starch using alkaline protease and neutral protease has also been reported. Zhao et al. [39] used response surface methodology to optimize the neutral protease extraction process of highland barley starch, and the optimal extraction conditions included an enzyme concentration of 140.79 U/g, an extraction temperature of 45.01 °C, and a hydrolysis time of 2.57 h. The enzymatic method can extract starch directly from seeds and retain most of the grain’s natural properties with a gentle separation process and a relatively smooth surface of the isolated starch granules. HBS extracted by enzymatic method has relatively rough surface with high gel hardness and gelatinization temperature [37].

The presence of β-glucan is a concern, as it absorbs a considerable quantity of water during washing and increases the slurry viscosity, making it difficult to separate in subsequent steps. Technology has been developed for separating high-purity starch from HB grains with different amylose contents. It is possible to separate starch and fiber fractions from whole barley flour in a semi-aqueous medium (50% ethanol) without affecting the viscosity of β-glucan. The majority of the starch isolates thus obtained had high purity large particles, with yields ranging from 22–39%. More significantly, the extraction efficiency of the β-glucan component was 77–90%, indicating that separation from the starch component during processing was efficient [40].

### 2.3. Combined Wet and Dry Extraction

Barley was subjected to milling and air classification operations to identify the graded fractions with high starch content based on particle size and composition, and the fractions were further separated based on the wet technique. The combination of wet and dry approaches can minimize water consumption and centrifugal load while increasing starch yield. During the milling and air classification procedures, most of the β-glucan is transferred to other components, which is advantageous for further separation and purification of starch.

## 3. Chemical Composition

There are significant differences in the chemical composition of HBS (Table 2). Amylose content varies among the eight starches from 22.72% to 26.90%. Amylose content of HBS varies due to the barley variety, climatic conditions, and soil type. The amylose content in a particle is proportional to its size; i.e., the smaller the particle, the lower the amylose concentration. Matveev et al. [41] showed that variations in amylose content result in differences in structural and thermodynamic features. The protein, lipid, phosphorus, and ash content of HBS vary according to the growing conditions and purity of the separated starch. Solvent extraction can eliminate lipids, but the amylose content of HBS will increase. The amount of protein residue in HBS produced via cellulase extraction is high. Yangcheng et al. [42] reported that phosphorus is present in HBS in the form of phospholipids.

## 4. Structure

### 4.1. Molecular Structure

#### 4.1.1. Molecular Weight Distribution

The rheological properties, retrogradation, gelatinization, and gel strength of starch are influenced by its molecular weight and distribution. Using gel permeation chromatography with multi-angle light scattering, Liu et al. [45] identified the relative molecular mass of HBS as 8.796 × 10^7^ g/mol. The molecular weight of HBS was greater than 4 × 10^7^ g/mol, and was mainly distributed as 4–6 × 10^7^ g/mol, 6–8 × 10^7^ g/mol, and >8 × 10^7^ g/mol fractions, with distribution ratios of 32.26%, 38.41%, and 29.33%, respectively. Naguleswaran et al. [46] reported that the weight average molecular weight of HB amylopectin is ~22.4 × 10^6^ g/mol, which is similar to the relative molecular mass of glutinous rice amylopectin [47]. Thus, the glutinous rice amylopectin can be used as a reference for the development and utilization of HBS.

#### 4.1.2. Chain Length Distribution

The chain length distribution (CLD) of HBS amylose (Am) and amylopectin (Ap) differs by genotype and depends on both genetics and environments. The measurement of the CLD of Am and Ap is usually performed using different techniques due to the limitations of each structural characterization technique. In general, fluorophore-assisted carbohydrate electrophoresis and high-performance anion-exchange chromatography are commonly used to measure the CLD of Ap. Using the HPSEC-MALLS-RI system, Naguleswaran et al. [46] observed that the average chain lengths of high-amylose, normal, and waxy HBS are 7.8, 14.5, and 107.3, respectively. According to a report by Czuchajowska et al. [48], the degree of polymerization (DP) of high-molecular-weight amylopectin in common HB amylopectin is higher than 35. Furthermore, the degree of polymerization of intermediate-molecular-weight and that of low-molecular-weight amylopectin is 15–35 and less than 15, respectively. Song et al. [49] demonstrated that the Ap molecules of high Am HBS have peak CLs similar to that of waxy and normal barley starch, but contains fewer short Ap branches with a DP of 6–12, and very short (DP 6–9) chains. Although Ap CLDs could be tested using the FACE and HPAEC, size-exclusion chromatography (SEC) is currently the primary technique utilized to obtain Am CLD [50]. As reported by You et al. [51], the number-average DP of high-Am HBS was 6000–7500. Even in the same barley genotype, the CLD of Am and Ap can differ depending on the granular structure of the extracted starch [52].

#### 4.1.3. Particle Size Distribution

The ratio of the number of particles to the total number of particles in different size ranges is known as the particle size distribution, and it has an impact on the function, application, and product quality of starch [53]. Granule sizes of starch granules vary according to the starch type, the growth region, and climate. Currently, the main instruments used to determine the size distribution of starch granules are scanning electron microscope (SEM) and laser particle analyzer (LPA). SEM has advantages in observing the morphological structure of starch granules, but has drawbacks in determining the particle size distribution, which requires numerical statistics of starch granule size in combination with micrographs, resulting in a relatively low accuracy of measurement. LPA can reflect the overall particle size distribution of starch with high accuracy, excellent reproducibility, simple operation and low cost. However, it is important to ensure that the sample has a good dispersion before the determination, otherwise the accuracy of the measurement will be affected [54]. The volume-average particle size of several types of HBS ranges from 4.70 to 23.17 μm, as shown in Table 3. A high amylose content is associated with lower average particle size. A and B-type starches have distinct properties, such as solubility, swelling power, viscosity, and gelatinization. Consequently, their application values are significantly different. A-type starch is a refined commercial starch that is commonly used in the food, chemical, and pharmaceutical industries, whereas B-type starch has low application value and is commonly used as a raw material for fermentation or in animal feed.

### 4.2. Particle Structure

The surface of HBS granules is smooth and partly uneven, with uniform size and shape distribution. The granule shape is mostly oval and disc-shaped, somewhat round, and polygonal [55]. The larger the granule diameter, the more spherical and regular the form, whereas granules with the smaller diameter granules have an irregular shape and contain a small quantity of protein and broken particles, and show agglomeration. The Maltese cross of starch granules in various HB types is close to the center of starch particles in an “X” shape. The strength of the Maltese cross is determined by particle size, relative crystallinity, and crystal orientation [43]. The particle shapes and sizes of HB starch are listed in Table 4. Figure 1 shows micrographs obtained using a scanning electron microscope (SEM), polarized light microscopy (PLM), and confocal laser scanning microscope (CLSM) from several HBS samples.

### 4.3. Molecular Structure

#### 4.3.1. Molecular Weight Distribution

Natural starch has three crystalline forms: A, B, and C. The A-type exists mainly in cereal starch, for example, in wheat, maize, and rice, whereas the B-type is found in tubers and amylose-rich crops, such as potatoes and bananas. C-type is primarily the crystalline form found in rhizome crops, such as beans. HBS has an A-type diffraction characteristic with prominent diffraction peaks at the diffraction angles 2θ of 15°, 17°, 18°, 20°, and 23° (Figure 2) [55]. However, the strength of the diffraction peak at 18° varies modestly across starch varieties. Waduge et al. [58] showed that amylose-lipid complexes created a V-type weak peak at 2θ = 20°, which is a common feature of barley starch. With increasing amylose content, the primary peak becomes weaker, but the peak centered at 2θ of 20° becomes progressively stronger. Starch X-ray diffraction (XRD) spectra usually feature sharp peaks and dispersive regions. On the basis of the total and amorphous areas of starch XRD spectra, the relative crystallinity can be calculated. The relative crystallinity of HBS was 10.72–43.21% in [43], 11.81–31.06% in [55], 20.8–21.9% in [42], and 20.3–23.9% in [59]. Compared with normal HBS, waxy HBS had a relative crystallinity of 33.0–37.1% [59], and the relative crystallinity of high-amylose HBS was 29.1% [40]. The proportion of crystalline regions, the size of the crystals, the direction of the double helices within the crystalline regions and the degree of their interaction, may play a role in the relative crystallinity of starches [43].

#### 4.3.2. Lamellar Structure

Small-angle X-ray scattering (SAXS) is commonly used to characterize the lamellar structure of starch granules. The amylopectin side chains are arrayed in parallel to form a double-helical crystalline region with a high electron cloud density, whereas the branched regions of amylopectin form an amorphous region in the lamellar structure. Together, these two regions form a lamellar structure of 9–10 nm. The characteristic peaks from the SAXS analysis of starch are formed because of the differences in electron cloud density in different regions of the layered structure. HBS has a prominent scattering peak at ~q = 0.6 nm^−1^, which corresponds to the crystalline-amorphous lamellar structure of the starch granule, and the thickness of the lamellar structure is around 9.33 nm [45,60].

#### 4.3.3. Ordered and Amorphous Structures

The crystal orderliness of starch granules includes the long-range orderliness of the neatly packed double helix and the short-range orderliness of the double helix packed at short distances from each other. The short-range ordered structure, as part of the long-range ordered structure, is a prerequisite for the existence of long-range ordered structures. Short-range ordered structures are less susceptible to disruption than long-range ordered structures, and their presence does not always imply the formation of long-range ordered structures [61]. The Fourier infrared spectrum is particularly sensitive to starch chain conformation and the presence of helical structures, and may quantitatively indicate the ratio of ordered and amorphous structures in starch. Wang et al. [60] reported that the degree of amorphous structures in HBS was 63.67%.

## 5. Physicochemical Properties

### 5.1. Gelatinization by Differential Scanning Calorimetry (DSC)

The gelatinization properties of starch are essential indicators of its quality. The gelatinization temperature of starch can be determined by DSC [62], thermomechanical analysis [63], nuclear magnetic resonance spectroscopy [64], and other approaches. DSC has become a simple and effective technology to study the gelatinization properties of starch due to the small amount of sample and fast testing speed. DSC is an analytical instrument for analyzing the relationship between the energy difference and temperature of the specimen and the reference material under the programmed temperature control. There are two types: compensated DSC and thermal flow DSC. The gelatinization temperatures and gelatinization enthalpy change (ΔH) of HBS were examined by DSC (Table 5). The higher the total starch content, the earlier the gelatinization onset time; however, a higher amylose concentration was associated with a later gelatinization onset time and higher gelatinization temperature. Starch granule size, amylose-lipid complexes, and amylopectin structure affect starch gelatinization properties [65]. Small particles have high onset temperature (To), peak temperature (Tp), conclusion temperature (Tc), and low enthalpy change (ΔH). The To shows a positive correlation with amylose content. The shorter chain content of the amylopectin molecule side branches is associated with a smaller encapsulated crystalline area, and a lower To, Tp, and ΔH of starch [66]. Amylopectin is the primary factor in the expansion of starch granules, and the amylose-lipid complex can hinder granule expansion. Compared to ordinary HBS, waxy HBS has a lower gelatinization temperature. As waxy starch contains less amylose, few amylose-lipid complexes are present. Compared to wheat starch, HBS has a lower gelatinization temperature and is easier to gelatinize. This may because wheat starch contains a large number of closely arranged small granular starches, which are harder to gelatinize and increase the paste-forming temperature. Knowing starch gelatinization properties is important for its application range.

### 5.2. Swelling Power and Solubility

The swelling power of starch is expressed as its capability to absorb water during gelatinization, and the paste’s capability to hold water after centrifugation [67]. The mass of starch dissolved at a certain temperature when it reaches saturation in 100 g of water is known as starch solubility. Both indicators can reflect the intensity of starch-water interactions. Li et al. [43] concluded that the swelling power and solubility of various HBS types differed at a certain temperature range (50–90 °C), but both properties tended to increase with temperature. In the 50–60 °C and 80–90 °C temperature ranges, the swelling power was greatly raised, whereas, in the 80–90 °C temperature range, the solubility was significantly increased (Table 6). During heating, at around the starch gelatinization temperature, the microcrystalline bundle structure loosens, polar groups in the starch are exposed, and moisture around the starch is rapidly absorbed, resulting in a rapid increase in the swelling power. The factors influencing the swelling power and solubility of starch granules include the content, ratio, molecular weight, and branching of amylose and amylopectin [68]. Li et al. [69] found that with the increase of amylose and fat content in starch, the swelling power of HBS reduced, whereas the solubility increased. Starch with a high amylose concentration has poor swelling power because during the heating process, small granular amylose is first exuded from the starch granules and forms a stable three-dimensional network structure on the outside, wrapping the swollen starch granules and affecting the expansion and decomposition of starch granules. β-glucan also affects the swelling power of starch. Li et al. [70] showed that waxy and normal barley β-glucans could greatly reduce barley starch granule swelling. Wheat starch has lower solubility and swelling power than HBS at the same temperature, and a greater association of solubility and swelling power of HBS with temperature is observed than that of wheat starch. This is because the latter contains more densely packed small granular starch, which hinders its water absorption and expansion capabilities, affecting the solubility and swelling power of wheat starch.

### 5.3. Rheological Properties

#### 5.3.1. Pasting

Rapid visco-analyzer (RVA), Brabender viscograph, and rheometer can be used to analyze the pasting characteristics. RVA is more commonly used technique due to the small sample size required and fast detection speed. Table 7 describes the pasting properties of different varieties of HBS. The peak viscosity (PV) is indicative of swelling of starch granules before disintegration [73]. Breakdown (BD) is a measure of PV and the trough viscosity (TV) and can be used to indicate the extent of granule disintegration [74]. During the cooling procedure, the final viscosity (FV) gradually increases, which represents the stability of the cooled–cooked paste. The difference between FV and PV is setback (SB), which indicates the retrogradation capacity of starch [74]. HBS showed lower PV and TV than wheat starch, but a higher BD, indicating a small extent of swelling of HBS granules, and the swollen starch had lower strength and was easily ruptured, the thermal paste became unstable as a result. The purity of starch, the size of the granules, amylose and amylopectin content, the amylose/amylopectin ratio, interactions and their stability of the double helices in granules are factors that influence pasting capabilities [43].

#### 5.3.2. Flow

Several raw materials used in food machining are fluids, and their rheological characteristics are modified by the temperature and concentration of the food material, which in turn affects processing properties, such as food output, mixing, and agitation. Cereal powders, such as starch, are frequently utilized as pastes, and the rheological properties of the paste can influence product qualities, including viscosity, hardness, and texture. Typical starch pastes exhibit non-Newtonian shear-thinning fluid characteristics [75]. Zhu et al. [76] showed that highland barley gel exhibits the shear-thinning characteristics and belongs to the non-Newtonian fluid group. The viscosity of HBS paste increased as the concentration of starch slurry and ion concentration are increased, and decreased as the temperature, the shear rate, and heating time are increased. In contrast to wheat starch, HBS is more vulnerable to these factors and has poorer shear resistance properties.

### 5.4. Retrogradation

#### 5.4.1. Freeze-Thaw Stability and Syneresis

During the process of freezing gelatinized or ungelatinized starch at low temperatures (e.g., −18 °C) and subsequently thawing it at normal or higher temperature (e.g., 30 °C) to melt the starch, the changing trend and degree of starch physicochemical properties and granular structure reflects the freeze-thaw stability of starch, which has a direct impact on the textural attributes of associated quick-frozen foods. The amylose content affects freeze-thaw stability, with high amylose content resulting in poor freeze-thaw stability. Studies have shown that after four freeze-thaw cycles, there was a 4% syneresis for zero amylose HBS, compared to 21% for CDC Candle HBS with 5% amylose [77]. Zheng et al. [78] found that after one freeze-thaw cycle, the net syneresis of 5% waxy corn starch paste was 24%; this increase was three times higher than that of CDC Candle and zero amylose HB starches. Therefore, compared to waxy corn and CDC Candle, zero amylose HB starch showed a high freeze-thaw stability. Zhang et al. [79] showed that the syneresis of HBS was 57.4%, and the gel was hard and prone to rupture and shear-thinning when pressed, whereas the gel of wheat starch was soft, with a syneresis of 61.5%. Therefore, the freeze-thaw stability of HBS was better. The starch paste used in frozen foods must be frozen at low temperature or be used after being frozen and thawed multiple times. If the freeze-thaw stability is poor, the colloidal structure of starch will be destroyed and free water will precipitate after freezing and thawing. Food will be unable to maintain its natural texture, and its quality will be affected. As a result, HB is appropriate for use as a thickener or filler in frozen foods.

#### 5.4.2. Transparency of the Starch Paste

The degree of light transmittance after starch is completely formed into a paste is called the transparency of starch paste, which can be expressed as light transmittance. Greater light transmittance correlates with higher transparency. The transparency of starch paste is positively associated with the amylopectin content [80]. The solubility of starch also affects its transparency. Starch with high solubility expands and gelatinizes more easily, and the starch paste is more transparent. Conversely, lower solubility is associated with worse transparency. At 0 h of storage, compared with wheat starch paste, HBS paste had a higher light transmittance (16.5%) [81]. With increasing storage time, the light transmittance of both HBS and wheat starch paste decreased [82]. The retrogradation speed of starch paste was fast in the early stage of storage, and the light transmittance of both starches declined quickly; however, as time passed, the retrogradation speed slowed and gradually became saturated, causing the transmittance to gradually reduce to the limit value [83]. Waxy HBS has good transparency owing to its high amylopectin content, and the transparency is unchanged even after storage or retrogradation [84].

#### 5.4.3. Gel Textural Properties

The viscoelasticity and gel strength affect the machining and molding of the gel, as well as its flavor and quality. The mechanical taste of starch gel changes the viscoelasticity, hardness, and roughness of food, which is different from the chemical taste caused by sugar, inorganic salts, acid, alkali, etc. Gel strength varies with temperature; however, this change is reversible. During storage, amylopectin recrystallization is the major cause of the increase in gel hardness [85]. The presence of amylose promotes the recrystallization of amylopectin but does not affect the final degree of amylopectin crystallization. The gel formed by wheat and buckwheat starch has high strength and is difficult to break, whereas the gel formed by HBS is comparatively brittle and easily broken when squeezed. In contrast to buckwheat starch gel and wheat starch gel, HBS gel has higher cohesion, proper hardness, chewiness, and resilience than buckwheat and wheat starch.

### 5.5. Digestibility

Based on the speed of starch digestion, starch is nutritionally classified as rapid digest starch (RDS), slow digest starch (SDS), and resistant starch (RS) [86]. Studies [87] have shown that the RDS content is positively correlated with the glycemic index and linked to diseases, such as diabetes, obesity, and vascular disease. SDS are starches that are digested at a slower rate, resulting in the slow release of blood glucose without a quick spike in blood glucose levels. RS is a dietary fiber that is not digested in the gut but fermented directly in the large intestine and is thought to be effective in preventing intestinal diseases, such as colon cancer.

HBS contained RS, SDS, and RDS within ranges from 13.1–31.6, 19.8–25.7, and 14.4–19.6 g/100 g dry starch, respectively [88]. Shen et al. [89] reported that the hydrolysis rate of buckwheat starch is 10–30%, which is significantly lower than that of oat starch (30–70%) and HBS (20–60%). HB RS exhibited higher hydrolysis rates than oat RS and buckwheat RS. Amylose and amylopectin have different susceptibilities to amylolytic enzymes, and amylopectin is less resistant to enzymatic hydrolysis [9]. Starch with a lower proportion of short-chain amylopectin is less prone to hydrolysis [90]. Normal and waxy HBS are less resistant to enzymatic hydrolysis compared to high-amylose HBS [91]. RDS decreases with increasing amylose concentration [9]. Starch digestibility is influenced by molecular weight, degree of branching, crystal type, crystallinity, surface properties, and size [36]. Small starch granules have higher digestibility than large starch granules [36]. Moza and Gujral [18] showed that the contents of RDS, SDS, and RS varied greatly with altitude, and SDS was positively correlated with altitude. The in vitro digestibility of HBS improves after removal of endogenous non-starch components [36].

HB contains an average of 5.25% β-glucan, which inhibits the functions of α-glucosidase, α-amylase, and invertase, thereby affecting the digestibility of HBS [92]. With higher concentration and molecular weights of β-glucan, the viscoelasticity of the β-glucan solution added, its inhibitory effect on α-amylase activity became stronger, and it became more efficient in delaying starch digestion. These findings suggest that the high viscosity of HB β-glucan solution may be a contributing factor to the low starch digestion of HB [93]. However, according to Zhang et al. [94], the reticular structure of β-glucan has a more profound effect on starch digestibility than the viscosity. Deng et al. [5] showed that waxy cookies contained higher protein and β-glucan contents and lower total starch content, and retained higher nutritional value and potential health benefits. Breads with a high proportion of hulless barley wholegrain flour have a higher nutritional value [95].

Using appropriate chemical, physical, and enzymatic modification techniques, the particle, lamellar, crystalline, amorphous, and chain structures of starch can be regulated to modulate starch digestibility and confer diverse nutritional benefits. Currently, the United States, Japan, Australia, and other countries have developed commercial anti-digestive starch products, some of which have up to 90% anti-digestibility properties [96].

## 6. Applications

Based on its unique structure, properties, and chemical composition, HBS is especially suited for use in food and non-food applications. The high water holding capacity and salt-resistant gelatinization stability of HBS make it useable in meat products and as a soup thickener. Shand [97] found that the addition of waxy starch hulless barley resulted in excellent water-holding capacity of low-fat pork bologna during storage. In contrast to buckwheat starch gel and wheat starch gel, HBS gel has higher cohesiveness, proper hardness, chewiness, and resilience than buckwheat and wheat starch, and can be used in producing gum-based confections, bread flour, etc. [98]. HBS has better freeze-thaw stability than other cereal and tuber starches. The shrinking of HBS due to dehydration is substantially lower than that of waxy corn starch, even after repeated freeze-thaw procedures. When the amylopectin content in HBS reaches 92% to 100%, it shows outstanding viscosity and freeze-thaw stability, and can be widely used as a thickener or filler for frozen foods [79]. Chang and Lv [71] reported that HBS has favorable emulsification stability and water-oil binding capacity. It can be widely used in moisturizing emulsions as an emulsification stabilizer; the structure of emulsions can be improved, the properties can be maintained, and chemicals used for thickening and emulsification can be greatly reduced. Several studies have reported that waxy HBS contains more amylopectin and β-glucan, and a higher β-glucan content makes waxy HBS indigestible. A study was performed by Izydorczyk et al. [99] on Asian noodles that were enriched with fiber components from roller milling of hull-less barley. During in vitro digestion of the noodles, the release of glucose was reduced, suggesting a possible reduction in the glycemic index and an increase in the nutritional benefits of the noodles. HBS has relatively high solubility and excellent transparency. In particular, the transparency of waxy HBS paste is better, even after refrigeration or retrogradation, and the transparency of the paste is unchanged [84]. Therefore, HBS can be used in clear juice drinks. Recent studies have shown that HBS has good film-forming properties, excellent biocompatibility, biodegradability, low immunogenicity, and no toxic side effects, making it ideal for embedding into core materials and retaining their physicochemical features [100]. HBS can be chemically modified to obtain a variety of properties that can expand its applications. Cross-linked HBS exhibits swelling resistance, high temperature tolerance and viscosity, making it an ideal ingredient for soups, gravies, sauces, etc. [101]. Oxidation promotes low retrogradation and viscosity, and these properties are important for the production of biodegradable films [102]. The acetylated HBS had higher RS content comparing to native starch, proving its nutritional value as a low glycemic index food [71]. Further research on HBS by natural and synthetic means is necessary to expand its application.

## 7. Conclusions

Our knowledge of the properties of HBS has progressed significantly. The main component of HB is starch, which accounts for 58.1–72.2% of its dry weight. The starch obtained by milling and air classification of barley, followed by wet separation, is of high purity. Several studies have shown significant differences in the chemical composition, particle and molecular structure, and physicochemical characteristics of HBS. In general, HBS granules have a smooth surface, and most are oval or disc-shaped. Normal, waxy, and high-amylose HBS have an A-type crystalline form with degrees of crystallinity ranging from 10.72–43.21%, 33.0–37.1%, and 29.1%, respectively. The pasting properties of distinct highland barley starches vary and are influenced by the purity of the starch, amylose/amylopectin ratio, amylose and amylopectin content, and granule size. HBS is widely used in food and other industries because of its outstanding freeze-thaw stability, high solubility, excellent emulsifying stability, and favorable stability. Non-starch components are key factors contributing to the low glycemic index of HB-based foods. The variability in starch structures and properties can be further enhanced by physical and chemical modifications. To maximize the use of HBS, it is necessary to consider the relationship between chemical composition, structure, properties, modifications and applications, not just from a one-sided perspective.

## Figures and Tables

**Figure 1 foods-12-00387-f001:**
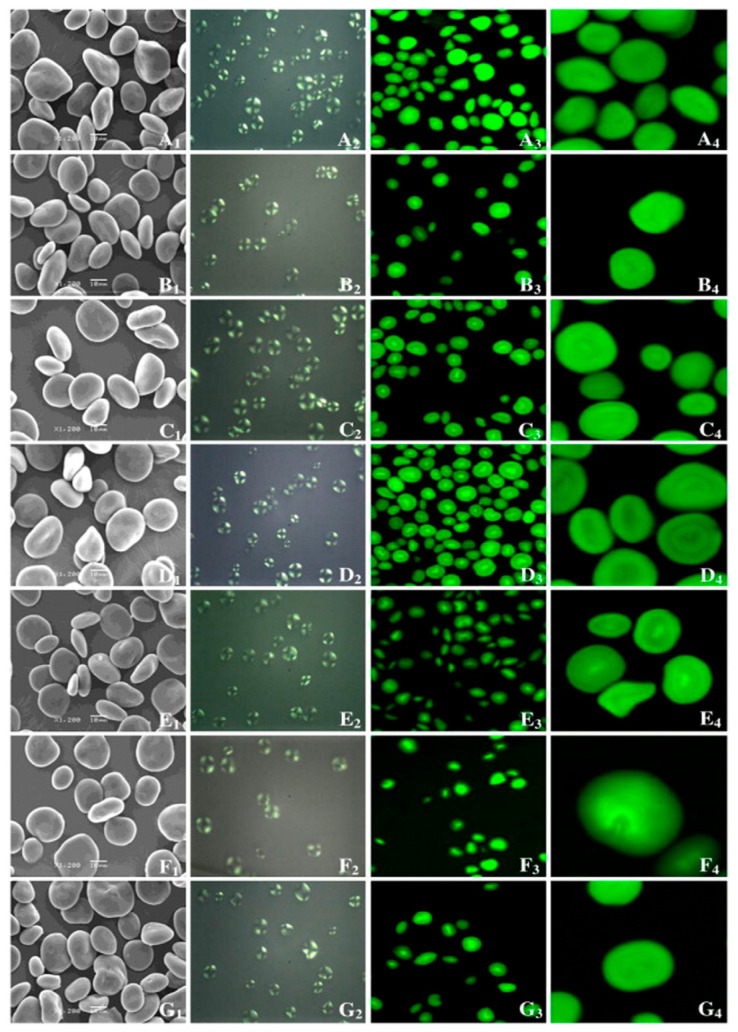
Scanning electron micrographs (SEM) (1); polarized light micrographs (PLM) (2); and confocal laser scanning micrographs (CLSM) (3, 4) of starches from seven highland barley cultivars. (**A**) Zangqing 8; (**B**) Zangqing 148; (**C**) Beiqing 6; (**D**) Zangqing 25; (**E**) Kunlun 12; (**F**) Zangqing 320; (**G**) Xila 19 [43].

**Figure 2 foods-12-00387-f002:**
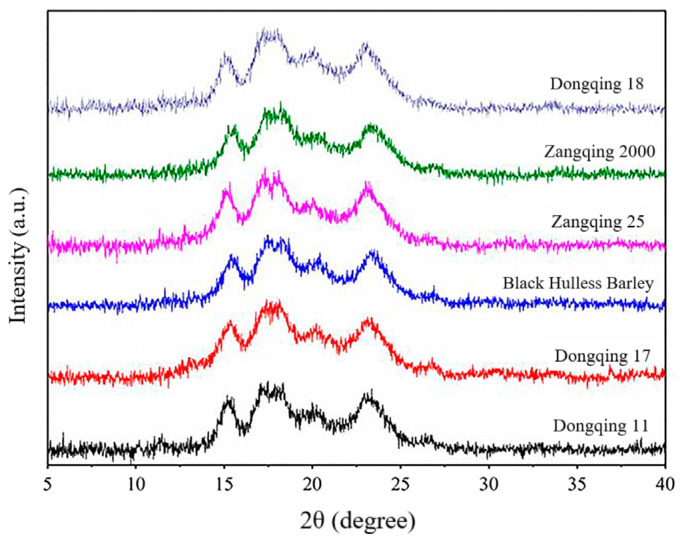
Typical X-ray diffraction spectra of starches isolated from six highland barley cultivars [55].

**Table 1 foods-12-00387-t001:** Composition of highland barley grain.

Nutrients	Summary of Research Results	References
Starch	HB had lower levels of starch (58.1–72.2%) than wheat (70–75%), corn (65–74%), and rice (~80%)	[9,10,11,12]
Protein	The protein content of HB was 8.20–20.80%, similar to wheat (8–20%), and higher than rice (6–7%) and corn (6–12%)	[13,14,15,16]
Lipid	The crude lipid content in HB was about 2.01–3.09%, which was higher than rice, but lower than corn, sorghum and oat	[16]
Fiber	HB contained 12.8–17.2% fibers, higher than most cereals, especially β-glucan	[17,18]
Mineral	The mineral content of HB was 1.46–2.20%, similar to normal staple foods, such as rice, wheat and corn	[19]
Vitamins	HB had about 39.0–379.7 mg/kg vitamin E, and 30.4–1327.4 mg/kg vitamin B, which was higher than the average of maize (3.9–36.3 mg/kg) and wheat (0.16–13.55 mg/kg)	[20,21,22]

**Table 2 foods-12-00387-t002:** Chemical composition of highland barley starch.

Varieties	Amylose (%)	Protein (%)	Lipid (%)	Phosphorus (%)	Ash (%)	References
Zangqing 8	23.85	0.42	0.02	- ^1^	-	[43]
Xila 19	22.72	-	0.01	-	-	
Kunlun 12	24.97	0.45	0.01	-	-	
Kangqing 3	26.90	-	0.42	0.047	-	[42]
Beiqing 7	24.80	-	0.45	0.048	-	
CDC McGwire	-	0.07	0.14	0.046	0.30	[40]
CDC Freedom	-	0.19	0.15	0.051	0.29	
CDC Dawn	25.80	-	-	-	-	[44]
Falcon	23.8	-	-	-	-	

^1^ -, the effect was not evaluated in the study.

**Table 3 foods-12-00387-t003:** Particle size distribution of highland barley starch.

Varieties	D(10) (μm)	D(50) (μm)	D(90) (μm)	References
Dongqing 11	- ^1^	18.99	-	[55]
Black HB	-	22.51	-	
Nakano blue 25	-	20.33	-	
Dongqing 18	-	23.17	-	
Beiqing 6	33.60	4.70	55.11	[48]
Dongqing 11	-	13.10	-	[42]

^1^ -, the effect was not evaluated in the study.

**Table 4 foods-12-00387-t004:** Morphology of highland barley starch granules.

Varieties	Size (μm)	Shape	Reference
Beiqing 4, Beiqing 6, Beiqing 7, Kangqing 3, Kangqing 6, and Kangqing 7	2.0–13.8	Lenticular, spherical	[42]
Zangqing 8, Zangqing 148, Beiqing 6, Zangqing 25, Kunlun 12, Zangqing 320, and Xila 19	10–30	Oval, disk-like, and irregular	[43]
CDC Alamo, CDC Candle, CDC Dawn, and Phoenix	6.2–9.8	Lenticular, oval, and irregular	[56]
Dongqing 18, Zangqing 2000, Zangqing 25, Black HB, Dongqing 17, and Dongqing 11	18.99–23.17	Oval, spherical, and polygonal	[55]
HRF, QK, YBL, and SX	2–25	Lenticular, oval, and disk-like	[57]

**Table 5 foods-12-00387-t005:** Gelatinization properties of highland barley starch by DSC.

Varieties	Starch: Water Ratio (*w*:*w*)	Scanning Rate (°C/min)	To ^1^ (°C)	Tp ^2^ (°C)	Tc ^3^ (°C)	△H ^4^ (J/g)	References
Zangqing 8	3:12	10	57.30	60.56	69.88	8.93	[43]
Zangqing 25	3:12	10	57.02	60.79	70.69	9.03	
Zangqing 148	3:12	10	55.03	57.84	65.49	7.74	
Zangqing 320	3:12	10	55.93	59.11	70.25	9.16	
Beiqing 6	3:12	10	58.47	61.54	71.86	9.82	
Kunlun 12	3:12	10	56.17	59.00	72.74	9.74	
Xila 19	3:12	10	54.07	57.51	69.92	9.16	
BQ 6	- ^5^	-	54.1	-	63.6	10.5	[42]
KQ 6	-	-	56.1	-	63.5	10.3	
Dongqing 18	2:7	10	61.25	67.57	82.77	8.92	[55]
Dongqing 11	2:7	10	58.73	65.12	83.36	9.84	
Dongqing 17	2:7	10	57.81	64.36	84.70	10.66	
Zangqing 2000	2:7	10	59.40	67.51	82.60	7.14	
Zangqing 25	2:7	10	58.87	65.13	82.03	8.66	
Black HB	2:7	10	58.65	66.34	82.94	8.56	
HRF (Qinghai)	5:15	10	53.7	58.5	64.3	10.3	[57]
SX (Shanxi)	5:15	10	57.7	61.8	66.0	9.5	
HB (Tibet)	2:6	5	54.0	58.0	62.1	10.4	
YX (Yunnan)	2:6	5	53.4	57.7	61.9	9.7	
Phoenix	2:6	5	53.1	59.1	71.0	12.8	[56]
CDC Dawn	2:6	5	52.0	58.1	72.5	12.7	

^1^ T_o_, onset temperature; ^2^ T_p_, peak temperature; ^3^ T_c_, conclusion temperature; ^4^ ΔH, enthalpy change; ^5^ -, the effect was not evaluated in the study.

**Table 6 foods-12-00387-t006:** Swelling power and solubility of highland barley starch.

Varieties	Parameters	Temperatures (°C)					References
		50	60	70	80	90	
Zangqing 8	SP ^1^	3.00	8.43	9.91	12.49	15.90	[43]
	S ^2^	1.21	4.26	3.68	7.62	18.63	
Beiqing 6	SP	2.55	8.40	10.91	12.33	17.35	
	S	0.59	2.19	4.37	6.64	19.55	
Kunlun 12	SP	3.21	8.94	10.72	12.35	15.67	
	S	0.59	2.00	2.53	5.06	17.42	
Xila 19	SP	3.74	8.51	9.44	11.31	13.33	
	S	1.78	3.34	3.49	5.83	17.31	
Linzhou 148	SP	3.55	7.46	9.88	11.87	13.23	
	S	2.62	5.77	9.49	15.15	15.71	
HB	SP	3.59	5.53	6.73	11.15	18.56	[71]
	S	- ^3^	-	-	-	-	
Dulihuang	SP	0.03	0.07	0.06	0.08	0.09	[72]
	S	0.34	1.10	1.70	2.77	5.80	

^1^ SP, swelling power (g/g); ^2^ S, solubility (%); ^3^ -, the effect was not evaluated in the study.

**Table 7 foods-12-00387-t007:** Swelling power and solubility of highland barley starch.

Varieties	PV ^1^ (cP)	TV ^2^ (cP)	BD ^3^ (cP)	FV ^4^ (cP)	SB ^5^ (cP)	PT ^6^ (°C)	References
Zangqing 8	3012	2253	759	3094	841	53.83	[43]
Zangqing 148	3351	2700	651	3665	964	57.25	
Kunlun 12	3362	2636	726	3469	833	84.35	
Xila 19	2977	2502	474	3231	729	50.33	
Dongqing 18	264	237	27	406	169	93.80	[55]
Dongqing 17	354	245	109	545	300	94.65	
Dongqing 11	460	383	77	731	348	93.10	
Zangqing 2000	380	305	75	638	333	95.50	
Black HB	523	357	166	831	474	93.10	
HRF (Qinghai)	206	- ^7^	68	-	146	82.5	[57]
SX (Shanxi)	298	-	152	-	193	76.3	
YX (Yunnan)	294	-	135	-	171	79.1	
HB (Tibet)	234	-	83	-	154	83.1	

^1^ PV, peak viscosity; ^2^ TV, trough viscosity; ^3^ BD, breakdown; ^4^ FV, final viscosity; ^5^ SB, setback; ^6^ PT, pasting temperature; ^7^ -, the effect was not evaluated in the study.

## Data Availability

Not applicable.

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
