# Peer review of "Highland Barley Starch: Structures, Properties, and Applications"

_foods, 2023, doi:10.3390/foods12020387_

Round 1

Reviewer 1 Report

The Highland barley starch review article has a pertinent sequence for the order of presentation of the topics, and the tables and figures are well described. However, in terms of content, I suggest that the following points be reviewed:

Line 24, what do you mean by "super", what other material do you compare it to?

Line 26, what is a promising profile? could they be more specific or use less subjective terms?

Line 60, when starting the topic of isolation methods with their respective subtopics, damaged starch is never mentioned, it should be included.

Lines 97 and 99, enzymatic method and enzymatic extraction refer to different terms? Could the authors please clarify

Line 120, about chemical composition, the term remarkable is not appropriate because when you check the table the differences are given by multiple factors. Perhaps significant differences would be appropriate.

In line 145, when the amylopectin chains that make up the starch are mentioned, there is a lack of information about them, it would be appropriate to place the distribution percentages according to their DP.

Line 318, the discussion should include a comparison with some of the most used starches in the industry, thus highlighting the advantages of HBS.

The applications section only presents 5 references; it would be advisable to increase the information of the section since the term "applications" is part of the title, but the information is scarce.

The conclusion is repetitive compared to the abstract and introduction of the article, please revise.

Thank you for your attention.

Author Response

Dear Reviewer:

Thank you for the reviewer’s comments concerning our manuscript entitled “Highland barley starch: structures, properties, and applications”. Those comments are all valuable and very helpful for revising and improving our paper, as well as the important guiding significance to our researches. We analyzed comments carefully and have made a correction which we hope meets with approval. Changes to the manuscript have been made in RED so they are distinct from the original text. The corrections in the paper and the responses to the reviewer’s comments are detailed in the attachment.

Best regards,   Author

Reviewer 2 Report

1. Sentence in line 20 needs to change to be different from line 48

2. Tabel 1 is lined up 

3. lines 86 to 89 need references

4. lines 90 to 92 need references

5. the sentence, "in contrast..............temperature" contradicts the sentence before it.

6. references 38 and 39 are not listed in the text.

7. line 138, this method needs a brief description for the benefit of the.

8. The method in section 4.1.3 needs a description.

9. References 46, 47, and 48 are not in the text

10. In Section 5.1, you need to add some more information about these methods.

11. Reference 59 is not in the text.

12. Reference 55 is missing from the text.

13. Line 357,  a reference is needed for this statement.

 14. There are more current references on starch swelling in the last 3 years or so, please add one or two of them to your paper.

Author Response

(The authors gave the same response as above.)
